# Transcriptional Dysregulations of Seven Non-Differentially Expressed Genes as Biomarkers of Metastatic Colon Cancer

**DOI:** 10.3390/genes14061138

**Published:** 2023-05-24

**Authors:** Xiaoying Lv, Xue Li, Shihong Chen, Gongyou Zhang, Kewei Li, Yueying Wang, Meiyu Duan, Fengfeng Zhou, Hongmei Liu

**Affiliations:** 1School of Biology and Engineering, Guizhou Medical University, Guiyang 550025, China; xiaoyinglv163@163.com (X.L.); lixuexue525@163.com (X.L.); chenshihong@stu.gmc.edu.cn (S.C.); zgy1943541699@163.com (G.Z.); 2Engineering Research Center of Medical Biotechnology, Guizhou Medical University, Guiyang 550025, China; 3School of Public Health, the Key Laboratory of Environmental Pollution Monitoring and Disease Control, Ministry of Education, Guizhou Medical University, Guiyang 550025, China; 4Key Laboratory of Symbolic Computation and Knowledge Engineering of Ministry of Education, Jilin University, Changchun 130012, China; kwbb1997@gmail.com (K.L.); wyy180320@163.com (Y.W.); dmy235813@163.com (M.D.); 5College of Computer Science and Technology, Jilin University, Changchun 130012, China

**Keywords:** metastatic colon cancer, mqTrans, dark biomarker, gene expression

## Abstract

**Background**: Colon cancer (CC) is common, and the mortality rate greatly increases as the disease progresses to the metastatic stage. Early detection of metastatic colon cancer (mCC) is crucial for reducing the mortality rate. Most previous studies have focused on the top-ranked differentially expressed transcriptomic biomarkers between mCC and primary CC while ignoring non-differentially expressed genes. **Results:** This study proposed that the complicated inter-feature correlations could be quantitatively formulated as a complementary transcriptomic view. We used a regression model to formulate the correlation between the expression levels of a messenger RNA (mRNA) and its regulatory transcription factors (TFs). The change between the predicted and real expression levels of a query mRNA was defined as the mqTrans value in the given sample, reflecting transcription regulatory changes compared with the model-training samples. A dark biomarker in mCC is defined as an mRNA gene that is non-differentially expressed in mCC but demonstrates mqTrans values significantly associated with mCC. This study detected seven dark biomarkers using 805 samples from three independent datasets. Evidence from the literature supports the role of some of these dark biomarkers. **Conclusions:** This study presented a complementary high-dimensional analysis procedure for transcriptome-based biomarker investigations with a case study on mCC.

## 1. Introduction

Colon cancer (CC) is a common digestive tract malignancy affecting both sexes. It is the fifth most frequently diagnosed cancer worldwide [1]. Many CC patients present with metastatic features at diagnosis due to negligible symptoms in the early stages [2]. Metastatic colon cancer (mCC) has a high recurrence rate even after administering treatment, such as colectomy [3]. These factors make CC the fifth deadliest tumor worldwide [1,4].

The mortality rate and prognosis of CC patients will be greatly improved if the disease can be diagnosed early [5]. However, the early symptoms of CC are often difficult to detect, and progression is rapid after metastasis. Stool-based and fecal immunochemistry tests are useful for non-invasive CC detection; however, these tests lack sensitivity and specificity for early CC lesions [6,7]. Invasive screening technologies, such as lower gastrointestinal endoscopy, can detect the early signs of large lesions depending on the practicing clinician’s experience [5,8]. Carcinoembryonic antigen (CEA) serves as a general serum biomarker for multiple solid tumors, and its combination with the levels of carbohydrate antigen 19-9 (CA19-9) has been used for the clinical diagnosis, therapy response, and prognosis follow-up decision of CC patients [9,10]. The rapid development of high-throughput transcriptome profiling technologies, such as RNA-seq [11] and microarray [12], enables the detection of molecular biomarkers for CC diagnostics and treatment planning.

Various statistical and machine-learning approaches have been designed to identify diagnostic and prognostic CC biomarkers. Hammad et al. evaluated the statistical significance of differentially expressed genes and determined those with *p*-values < 0.05 and log2(fold-change) > 1.5 to be colorectal cancer biomarkers [13]. Zhang et al. combined statistical differential expression analyses and machine-learning approaches to detect potential CC biomarkers, and a further investigation into differential protein levels confirmed that Chromogranin A (CHGA) served as an ideal diagnostic biomarker for CC [14]. Qin et al. proposed a ratio-based feature engineering rule to transform the miRNA expression landscape into the pairwise ratio space and obtained a CC diagnostic model that was stably validated across multiple datasets and profiling platforms [15]. Furthermore, support vector machine (SVM) analysis has been used to identify a 15-gene signature for detecting CC recurrence [16].

This study hypothesized that the collaborative regulation of multiple transcription factors (TFs) on messenger RNAs (mRNAs) could be quantified to illustrate the transcription regulatory landscape of a transcriptome. TFs are indispensable proteins in the eukaryotic transcription regulatory machinery that closely participate in tumor microenvironment cellular functions [17], including cancer cell proliferation, migration, and invasion [18,19]. The regulation of multiple TFs for an mRNA was quantified as a regression model from the expressions of TFs to that of the mRNA [20,21,22]. The difference between the predicted and real expression level of an mRNA can be calculated as the mqTrans value of the specific mRNA in a query sample [23]. The mqTrans space of a query sample quantitatively reflects the transcription regulation changes in that sample versus the training samples in regression models. We defined a dark biomarker in mCC as an mRNA gene differentially expressed in the mqTrans space but non-differentially expressed in the original gene expression levels. We called these genes dark biomarkers, because most existing approaches would disregard these genes due to their non-differential expression levels. However, the significantly altered mqTrans values suggest that these biomarkers undergo transcriptional dysregulation in mCC. The identified dark biomarkers are potential novel diagnostic and therapeutic targets that can be experimentally investigated by assessing overlapping long non-coding RNAs (lncRNAs), encoded proteins, and interacting competing endogenous RNAs (ceRNAs).

## 2. Materials and Methods

### 2.1. Datasets

Three transcriptomic CC datasets were retrieved from the Gene Expression Omnibus (GEO) database (https://www.ncbi.nlm.nih.gov/gds/, last accessed on 10 June 2022) [24] (Table 1). The GEO GPL570 platform was used to profile the expression levels of 54,675 probe sets, and the detailed annotations of the samples and features were collected from the GEO database. We curated the metastasis annotation based on the TNM staging system, where the M parameter defined the metastasis status [25]. A sample was annotated as mCC if the M parameter exceeded 0, if the metastasis date was available, or if the sample diagnosis indicated a distant metastatic organ. All other samples were denoted as primary CC.

### 2.2. Preprocessing Procedure

A preprocessing procedure was applied to the three datasets, and 25 samples in the GSE39582 dataset without clearly labeled metastasis information were removed. The annotation file of the transcriptome profiling platform GPL570 was retrieved from the GEO database [24]. Transcriptomic features were denoted as TF features if their corresponding gene symbols were transcription factors according to the AnimalTFDB 3.0 database (http://bioinfo.life.hust.edu.cn/AnimalTFDB/#!/, last accessed on 23 March 2022) [29]. Features with corresponding mRNA gene symbols were regarded as mRNA features.

### 2.3. Design of Experiment

The GSE39582 dataset is much larger than the other two datasets and was split into the training and testing sub-datasets (Figure 1). The training sub-dataset comprises 60% of the randomly selected primary CC samples from the GSE39582 dataset. The remaining 40% of these primary samples and all of the metastatic CC samples from the GSE39582 dataset constitute the testing sub-dataset. The other two datasets (GSE3789 and GSE26906) were used to independently confirm the detected dark biomarkers. We also evaluated the mqTrans workflow using part of the training sub-dataset.

This study focused on the transcriptomic features that were non-differentially expressed in the original expression level data (*p*-value > 0.05) but differentially expressed after mqTrans value calculation (mqTrans *p*-value < 0.05). The significance of the differential expressions was evaluated using the statistical unpaired *t*-test function provided in the Python programming language version 3.8.3. The two sample groups comprised primary and metastatic CC patients, and the identified transcriptomic features were defined as dark biomarkers of mCC.

### 2.4. Functional Characterizations

The dark biomarker networks and interacting proteins were retrieved from the Search Tool for the Retrieval of Interacting Genes/Proteins (STRING) database (https://cn.string-db.org/, last accessed on 26 October 2022) [30] and visualized using Cytoscape software version 3.9.1 [31]. Functional annotations of the detected dark biomarkers were curated from the GeneCards database (https://www.genecards.org/, last accessed on 26 October 2022) [32].

Supporting literature was screened in the PubMed database (https://pubmed.ncbi.nlm.nih.gov/, last accessed on 26 October 2022) [33]. Since the dark biomarkers did not show differential expression in CC, we expanded the screening range to colorectal cancer (CRC), comprising colon and rectal cancer [34].

### 2.5. Screening lncRNAs with Overlapping Dark Biomarkers

We investigated the long non-coding RNAs (lncRNAs) overlapping the detected dark biomarkers and how they may facilitate the observed patterns of the dark biomarkers’ non-differential expressions in mCC. The lncRNAs were collected from the LncBook database 2.0 (https://ngdc.cncb.ac.cn/lncbook/home/, last accessed on 14 September 2022) [35]. There were 95,243 lncRNA genes downloaded. The chromosomal location information annotated by the detected dark biomarkers was compared, and lncRNAs with complete or partial overlaps with biomarker genes were screened and analyzed.

### 2.6. Detection of mCC Dark Biomarkers

There were 3501 transcriptomic features with corresponding gene symbols annotated as transcription factors (TFs) in the AnimalTFDB 3.0 database [29]. Linear regression models in the sklearn package were trained on the training primary cancer samples in the GSE39582 dataset using the mqTrans protocol [23,36]. We obtained stably converged regression models for 8313 mRNA features with PCC >0.5 between the predicted and real expression levels of a given mRNA feature. Each regression model produced at least one TF feature with non-zero weight.

The trained regression models were evaluated on the testing samples in the GSE39582 dataset, and 712 dark biomarkers were detected that satisfied the definition of non-differential expression in the original analysis (*p*-value > 0.05) but differential expression after mqTrans value calculation (mqTrans *p*-value < 0.05). The same criterion was used to screen the two independent datasets, GSE37892 and GSE26906, in which we detected 472 and 397 dark biomarkers, respectively. The overlapping dark biomarkers are shown as a Venn diagram in Figure 2a.

We further investigated the dark biomarker detection procedure using different numbers of training samples (Figure 2a). We denoted the list of seven dark biomarkers identified above as “Main.” Subsequently, we randomly extracted 50%, 40%, and 20% of primary cancer samples from the GSE39582 dataset as the training set (Figure 1) and determined the overlapping dark biomarkers in all three datasets (“Branch1,” “Branch2,” and “Branch3” in GSE39582, GSE37892, and GSE26906, respectively). As shown in Figure 2b, some dark biomarkers were stably detected after training the regression models with as low as 20% of the primary cancer samples.

## 3. Results and Discussion

### 3.1. Characterization of mCC Dark Biomarkers

We checked the co-existence of the gene symbols and colon/colorectal cancer in the title and abstract of the PubMed literature [33]. YTHDC2 has been reported to be an m6A RNA methylation regulator, but its expression level in colon adenocarcinoma was not significantly different from that in the adjacent mucosa, similar to that observed with other m6A RNA methylation regulators [37]. Tanabe et al. observed that the protein level of YTHDC2 was positively correlated with CC tumor stage and metastasis [38]. The dark biomarker gene LPP may be involved in regulating the proliferation, migration, and invasion of CRC cells by interacting with microRNAs [39]. Another dark biomarker gene, GIMAP1, was upregulated only in CD133-positive versus CD133-negative CRC stem cells [40].

Three of the identified dark biomarkers have not been investigated directly in CC or CRC (Table 2). The biomarkers 206339_at (CARTPT) and 222127_s_at (EXOC1) were found to be involved in tumor inflammation [41] and cell migration [42], and 213212_x_at corresponded to multiple genes, GOLGA6L4, GOLGA6L5P, GOLGA6L9, and LOC10272409, potentially because these paralogs share similar sequences [32,43]. The regression model of 213212_x_at achieved a PCC value 0.8275, supporting the precise prediction of its expression level. The original expression level of 213212_x_at was not significantly associated with the mCC phenotype (*p*-value = 0.7722); however, the mqTrans value showed significantly differential expression (*p*-value = 0.0022), suggesting strong transcriptional dysregulation of this dark biomarker.

We further visualized the distributions of the seven dark biomarkers as box plots in Figure 3. Each neighboring pair of box plots compares one dark biomarker in the metastatic (M, solid-filled) and primary (P, striped lines) colon cancer samples. The mqTrans values of the seven dark biomarkers differed significantly between the groups of metastatic and primary colon cancers in the GSE39582 dataset (Figure 3a), whereas their original expression levels were essentially the same (Figure 3b). The same patterns were observed in the other two datasets (Figure 3c–f).

Thus, the data reveal that the seven detected dark biomarkers demonstrate transcriptional dysregulation but unchanged original expression levels.

### 3.2. Dark Biomarkers Supported by Fewer Training Samples

Biomedical omics data exhibit the “large p small n” paradigm [45], i.e., the number of features is usually much larger than the number of samples. This is mainly caused by the difficulty of sample collection and the high cost of generating the omics profiles. Thus, we conducted an additional experiment to detect dark biomarkers using fewer training samples, i.e., 50%, 40%, and 20% of the primary colon cancer samples in the GSE39582 dataset. To maintain a fair comparison of the detected dark biomarkers, we kept the same testing sub-dataset of GSE39582 and extracted the 50%, 40%, and 20% training samples from the training set comprising 60% of the primary colon cancers in GSE39582.

As shown in Table 3, multiple dark biomarkers were confirmed using fewer training samples. The regression models confirmed three dark biomarkers (206339_at (*CARTPT*), 222127_s_at (*EXOC1*), and 241879_at (*LPP*)) with training using 50% of the primary colon cancer samples. The biomarkers 222127_s_at (*EXOC1*) and 241879_at (*LPP*) were confirmed using 40% of the training samples. Finally, 222127_s_at (*EXOC1*) could be stably recovered even using only 20% of the primary colon cancer samples in model training.

*EXOC1* is the gene exocyst complex component 1 and is involved in cell migration, similar to Sections 5 and 8 [42]. No literature was found that supports the connection between *EXOC1* and CC; however, genetic mutations in *EXOC1* have been associated with breast and cervical cancer [46,47]. Combining our experimental data and the literature evidence, *EXOC1* may be inferred as a dark biomarker of mCC due to its involvement in cell migration processes, and transcriptomics may not be the ideal technology to investigate its functional roles.

### 3.3. Many lncRNAs Overlap with Dark Biomarkers

LncRNAs are genomic regions encoding non-protein-coding RNA, typically >200 bp in length [48]. Multiple essential roles of lncRNAs have been investigated in the growth and metastasis of CC [49]. The genomic regions of many lncRNAs overlap with protein-coding genes, and the regulatory roles are exerted through the shared nucleotide sequences. Microarray- and sequencing-based transcriptome profiling technologies cannot easily discriminate whether transcripts from the same genomic region are encoded by mRNA or the overlapping lncRNA.

We hypothesized that overlapping lncRNA transcripts could have contributed to the miscalculation of the dark biomarker expression level (Table 4). Four dark biomarkers overlapped with lncRNAs. The dark biomarker gene LIM domain containing the preferred translocation partner in lipoma (LPP) overlaps with 15 lncRNAs, 10 sense and 5 antisense. Two LPP-overlapping lncRNAs, HSALNG0031464 and HSALNG0031470, showed differential expression levels under various physiological conditions according to LncBook database 2.0 data [35]. The upregulation of another LPP-overlapping lncRNA, HSALNG0031463 (LPP-AS2), promoted the proliferation and invasion of glioma cells in glioma tissues [50]. Similar observations were also found for the GOLGA6L5P-overlapping lncRNA HSALNG0107792 and the GOLGA6L9-overlapping lncRNA HSALNG0144423 [35]. This evidence suggests that the non-differential expressions of these dark biomarkers could have been compromised by the transcripts of their overlapping lncRNAs. LPP overlaps another antisense lncRNA, HSALNG0031469 (LPP-AS1), which is transcriptionally regulated by a celiac-disease-associated single-nucleotide polymorphism [51] and upregulated in the competing endogenous RNA network associated with the prognosis of human colon adenocarcinoma [52].

### 3.4. The Protein Level of YTHDC2 Is Associated with mCC

The dark biomarker *YTHDC2* encodes the protein YTH Domain Containing 2, an RNA modification 6-methyladenine (m6A) reader [53]. m6A is a common RNA methylation modification [37], requiring a methyl transferase (writer), demethylase (eraser), and methylation reading protein (reader) [54,55]. This modification is actively involved in RNA cleavage and translation and has recently been demonstrated to be associated with tumor growth, proliferation, and prognosis [37].

Previous studies have indicated the dark biomarker patterns of *YTHDC2*. *YTHDC2* is located on the long arm of human chromosome 5, and the subcellular localizations of its encoded protein include the cytoplasmic and perinuclear regions [44]. Liu et al. analyzed 13 m6A RNA methylation regulators using The Cancer Genome Atlas database data [56] and detected the abnormal expression of all of these regulators except YTHDC2 [37]. A later study demonstrated through Western blotting that the protein level of YTHDC2 was positively correlated with the stage and lymph node metastasis status of human gastric cancer cells [57]. The knockdown of *YTHDC2* changed the protein level of YAP (yes-associated protein) and significantly reduced the number of metastatic nodules. Tanabe et al. also confirmed through immunohistochemistry technology that the protein level of YTHDC2 is positively related to the staging, transfer, and invasion of CC [38].

In summary, *YTHDC2* did not show differential expression in mCC; however, the level of its encoded protein, YTHDC2, has been significantly associated with metastatic gastric cancer and CC.

### 3.5. mCC Association with Dark Biomarkers Based on the Protein–Protein Interaction Network

We manually curated the protein–protein interaction (PPI) network from the STRING database [30] to investigate the potential roles of the dark biomarkers in colon cancer. LIM And SH3 Protein 1 (LASP1) binds to the domain of the dark biomarker LPP and induces the proliferation and invasiveness of CRC via Hippo signaling and epithelial-mesenchymal transition (EMT) pathways [58,59]. The evolutionary conserved Hippo signaling pathway is involved in organ growth, and its dysregulation is frequently observed to interplay with the EMT pathway by promoting cancer chemoresistance and metastasis [60]. The crosstalk of the Hippo signals and EMT pathways is closely associated with the metastasis and prognosis of CRC. For example, the circulating lncRNA SNHG11 was observed to promote the CRC invasion and metastasis by targeting the Hippo signaling pathway and then promoting the EMT pathway [61]. Leptin (LEP) increases the recruitment of hypothalamic CREB Regulated Transcription Coactivator 1 (CRTC1) to the promoters of the KiSS-1 Metastasis Suppressor (KISS1) and the dark biomarker CARTPT [62]. The over-expression of KISS1 reduces the invasion capabilities of CRC cells by blocking the PI3K/Akt/NF-κB signaling pathway [63]. The PI3K/Akt/mTOR signaling pathway has been extensively investigated as a therapeutic target for CRC [64].

Rho Guanine Nucleotide Exchange Factor 12 (ARHGEF12, also known as LARG) is a tumor suppressor in breast and colorectal cancer cells. ARHGEF12 expression has been found to be downregulated in primary breast and colorectal cancer samples versus the normal tissue controls [65], and its encoded protein stimulates Rho signaling activation to promote CRC metastasis and liver invasion [66].

Thus, the PPI network in Figure 4 suggests that some of the identified dark biomarkers may exert their functions in mCC via their encoded or interacting proteins, although all these dark biomarkers do not show expression associations with mCC.

## 4. Conclusions

This study detected seven dark biomarkers of metastatic colon cancer using the mqTrans analysis protocol. The experimental data showed that these seven dark biomarkers showed non-differential expression between primary and metastatic CC across three independent datasets, and the engineered mqTrans values showed differential associations with mCC. The mqTrans value of an mRNA gene reflects the quantitative change in its transcription regulation, and the differential protein levels of some dark biomarkers in mCC were supported by the literature. The dark biomarkers may also exert functions in mCC via their interacting proteins.

The underlying mechanisms of the dark biomarkers remain to be resolved in future studies. LncRNAs overlapping mRNAs could be candidate causative factors in the miscalculation of transcript expression levels. Full-length transcript sequencing and quantitative protein identification technologies may be utilized to investigate the functional roles of the seven mCC dark biomarkers.

Due to the methodology limitation, the regression model quantitatively describes the collaborative effects of the regulatory TFs on a given mRNA gene and may not be directly used to measure the regulatory contribution of the individual TFs. Ablation-experiment-based evaluation and explainable artificial-intelligence-based regression will be conducted to interpret the scientific messages and insights conveyed from the TFs to their regulated mRNA gene in the future work.

## Figures and Tables

**Figure 1 genes-14-01138-f001:**
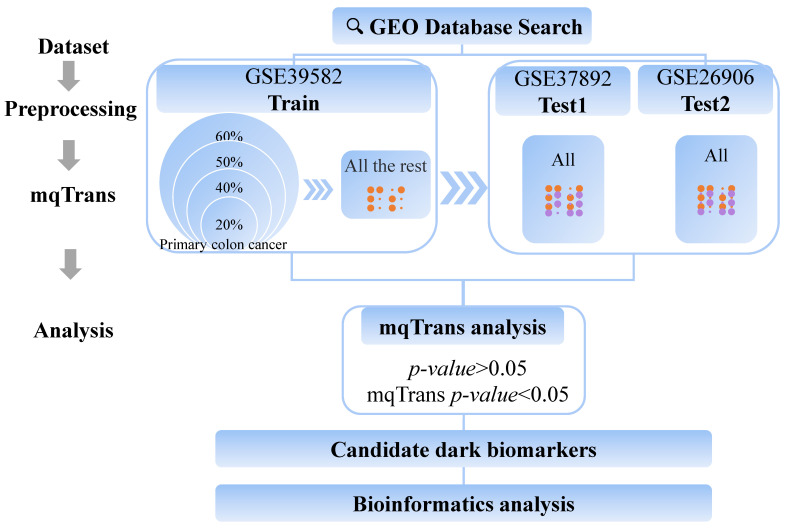
**Experimental workflow of this study**.

**Figure 2 genes-14-01138-f002:**
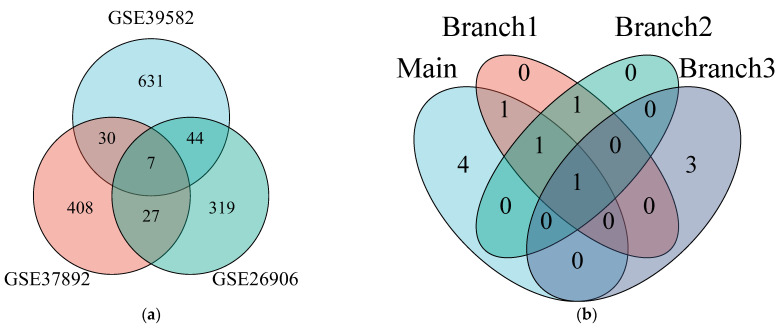
**Venn plots of the overlapping dark biomarkers in the three datasets.** (**a**) The dark biomarkers were detected by the regression models trained using 60% of the primary cancer samples from the GSE39582 dataset, and 7 dark biomarkers overlapped in the three datasets (Main). (**b**) We also evaluated dark biomarker detection using 50% (Branch1), 40% (Branch2), and 20% (Branch3) of the primary cancer samples to train the regression models. The Venn plots were generated using the package ggplot2 version 3.3.3 and the R programming language version 3.6.3.

**Figure 3 genes-14-01138-f003:**
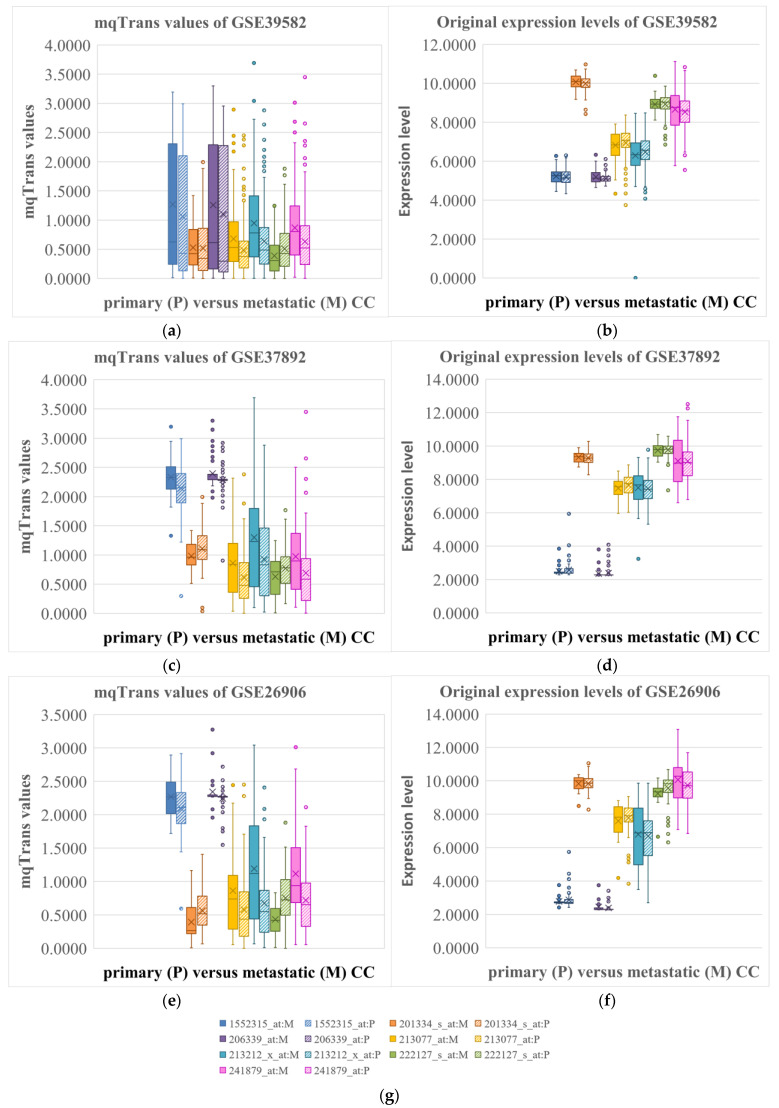
**Expression levels and mqTrans values of the seven dark biomarker features in the three datasets.** Box plots of the (**a**) mqTrans values and (**b**) expression levels of these seven dark biomarkers in the GSE39582 dataset are illustrated. (**c**) The mqTrans values and (**d**) expression levels in the GSE37892 dataset are also shown as box plots. The independent confirmations of the (**e**) mqTrans values and (**f**) original expression levels of these dark biomarkers in the GSE26906 dataset are also illustrated. (**g**) The values of each dark biomarker in the metastatic (M) and primary (P) colon cancer samples are shown in the solid and striped-line box plots with the same colors, respectively. The box plots are generated using EXCEL.

**Figure 4 genes-14-01138-f004:**
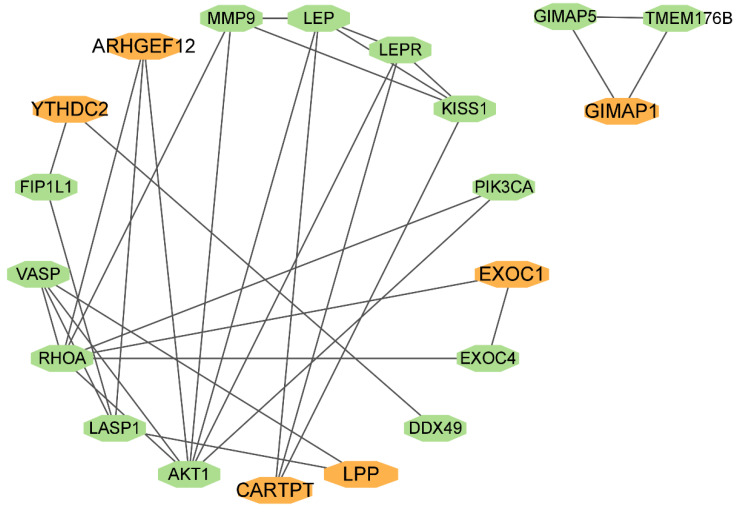
**Protein–protein interaction network involving six of the seven dark biomarkers.** The interaction annotations were collected from the STRING database [30]. The network illustration was generated using the Cytoscape software version 3.9.1.

**Table 1 genes-14-01138-t001:** **Summary of the three datasets used in this study.** The three datasets were profiled using the Affymetrix GeneChip Human Genome U133 Plus 2.0 (platform GPL570) transcriptome array. The first column is the annotation of each dataset. Each dataset contains different sample numbers but the same number of features. The numbers of primary (P) and metastatic (M) cancer samples are shown in the “Summary” column.

Notation	Accession	Samples	Features	Summary	Literature
Train	GSE39582	585	54,675	499 P vs. 61 M	[26]
Test1	GSE37892	130	54,675	93 P vs. 37 M	[27]
Test2	GSE26906	90	54,675	69 P vs. 21 M	[28]

**Table 2 genes-14-01138-t002:** **Dark biomarkers of metastatic colon cancer were confirmed in three independent datasets.** The “Dark biomarker” column lists the detected dark biomarkers as transcriptomic feature names. “Gene,” “Location,” and “Annotation” columns show the corresponding gene symbol, genomic location, and function annotations. The annotation data were collected from the GeneCards (https://www.genecards.org/, last accessed on 26 October 2022) [32] and UniProtKB (https://www.uniprot.org/, last accessed on 26 October 2022) [44] databases.

Dark Biomarker	Gene	Location	Annotation
206339_at	CARTPT	Chr5q13.2	CART prepropeptide
213077_at	YTHDC2	Chr5q22.2	YTH domain containing 2
213212_x_at	GOLGA6L4///GOLGA6L5P///GOLGA6L9///LOC102724093	Chr15q25.2	golgin A6 family-like 4/// golgin A6 family-like 5, pseudogene/// putative golgin subfamily A member 6-like protein 4-like/// golgin A6 family-like 9
222127_s_at	EXOC1	Chr4q12	exocyst complex component 1
241879_at	LPP	Chr3q27.3-Chr3q28	LIM domain containing preferred translocation partner in lipoma
1552315_at	GIMAP1	Chr7q36.1	GTPase, IMAP family member 1
201334_s_at	ARHGEF12	Chr11q23.3	Rho guanine nucleotide exchange factor (GEF) 12

**Table 3 genes-14-01138-t003:** **Dark biomarkers detected by fewer training samples.** The bold text indicates the dark biomarkers overlapping with the above section.

Train	Percentage of Primary CC Samples in GSE39582			
	60%	50%	40%	20%
Dark biomarkers	1552315_at	**206339_at**	209539_at	213911_s_at
	201334_s_at	209539_at	**222127_s_at**	218963_s_at
	**206339_at**	**222127_s_at**	**241879_at**	**222127_s_at**
	213077_at	**241879_at**		225575_at
	213212_x_at			
	**222127_s_at**			
	**241879_at**			

**Table 4 genes-14-01138-t004:** **Summary of the lncRNAs overlapping with the seven detected dark biomarkers.** The first two columns, “DB” and “Gene,” represent the dark biomarkers and their annotated genes. The dark biomarker 213212_x_at was annotated with four related genes. The next four columns, “Chr,” “Start,” “End,” and “S,” represent the specific position information of each gene on the human genome version 38. The last two columns provide the number of overlapping lncRNAs on the sense and antisense strands based on the LncBook database 2.0 [35].

DB	Gene	Chr	Start	End	S	Sense	Antisense
241879_at	LPP	chr3	188153284	188890671	+	10	5
222127_s_at	EXOC1	chr4	55853648	55905086	+	0	1
1552315_at	GIMAP1	chr7	150716606	150724284	+	1	0
213212_x_at	GOLGA6L4	chr15	84235773	84245358	+	1	1
213212_x_at	GOLGA6L5P	chr15	84507885	84516814	-	0	1
213212_x_at	GOLGA6L9	chr15	82429816	82439153	+	1	1
213212_x_at	LOC102724093	chr15	84350122	84359422	+	1	0
206339_at	CARTPT	chr5	71719275	71721048	+	0	0
213077_at	YTHDC2	chr5	113513694	113595285	+	0	0
201334_s_at	ARHGEF12	chr11	120336413	120489937	+	0	0

## Data Availability

All the datasets used in this study are publicly available, and their details are described in the section datasets. Gene Expression Omnibus (GEO) database (https://www.ncbi.nlm.nih.gov/gds/), AnimalTFDB version 3.0 database (http://bioinfo.life.hust.edu.cn/AnimalTFDB/#!/), STRING database (https://cn.string-db.org/), GeneCards database (https://www.genecards.org/), PubMed database (https://pubmed.ncbi.nlm.nih.gov/), UniProtKB (https://www.uniprot.org/) and LncBook database 2.0 (https://ngdc.cncb.ac.cn/lncbook/home/), accessed on 21 May 2023.

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
