# Peer review of "Transcriptional Dysregulations of Seven Non-Differentially Expressed Genes as Biomarkers of Metastatic Colon Cancer"

_genes, 2023, doi:10.3390/genes14061138_

Round 1
Reviewer 1 Report
Title
Dark biomarkers of metastatic colon cancer.
1. The mining of Dark biomarkers didn’t give enough information about the manuscript also “dark” is not a scientific adjective I suggest changing the title will give more value to the study and more attraction to readers using scientific keywords.
ABSTRACT
1. The English must be improved.
2. Concerning the abbreviation of metastatic colon cancer (MCC) it is known to be mCC.
Keywords
1. Did the authors mean differential expression by undifferential expression if so it must be corrected if not it shouldn’t be in the keywords as it is not a common scientific term it can be changed to gene expression as well.
INTRODUCTION
1. In this sentence of the manuscript, “Gut microbiome and carcinoembryonic antigen (CAE) have gradually become targeted CC biomarkers of a sub-population of the CC patients” the authors did not present the results of the references they listed first microbiome profile is still not a biomarker for CC and not used in diagnostic at all, on the other hand, CAE is general serum marker for most of the solid tumors in CC case clinicians use CEA with CA19 levels for CC diagnostic, therapy response and follow up.
2. Authors must define dark biomarkers in the introduction part and from where the nomenclature was given.
3. English must be improved.
MATERIALS AND METHODS
1. The last date of any database and webpage access must be added with the link.
2. A detailed description of the statistical model used to analyze the data must be added also the software used.
RESULTS & DISCUSSION
1. Figures quality and colors can be improved.
2. Most of the paragraph “Detection of MCC dark biomarkers” presents methods and the study design which must be switched to the methods section.
3. There is no clear scientific message between involved genes in mCC and the transcription factors the pathway presented in the results hasn’t been discussed on the prognostic value of mCC and compared to other similar findings.
4. Results and discussion must be reorganized depending on the purpose of the study.
CONCLUSION
It must be improved we usually present in the conclusion section a short summary of the study funding in this manuscript it will be better if the authors can make their conclusion shorter by only referring to their important results.
Authors must have a native English speaker better if not a scientist to review the manuscript or submit it for online English editing services to improve the quality of the manuscript and give better scientific soundness.
Author Response
May 18, 2023
Dear Editor of Genes,
Please find the revised manuscript “Transcriptional dysregulations of seven non-differentially ex-pressed genes as biomarkers of metastatic colon cancer” submitted for the journal Genes. The original submission ID is genes-2363380, and the title has been changed according to the reviewing suggestion. The manuscript has been revised and the point-by-point responses are provided to all the reviewing comments from the reviewers. All the revisions are highlighted in red in the revised manuscript.
We really appreciate the insightful comments from the two anonymous reviewers that have help improve the manuscript a lot. We indicate our appreciation in the section Acknowledgements.
We hope that our revisions satisfied the requests of the editor and anonymous reviewers.
Thank you for giving us the revising opportunity to improve our manuscript!
Sincerely,
Fengfeng Zhou, Ph.D., Professor of Health Informatics
College of Computer Science and Technology, Jilin University.
Email: FengfengZhou@gmail.com or ffzhou@jlu.edu.cn
Reviewer 1, Question 1
Title
Dark biomarkers of metastatic colon cancer.
- The mining of Dark biomarkers didn’t give enough information about the manuscript also “dark” is not a scientific adjective I suggest changing the title will give more value to the study and more attraction to readers using scientific keywords.
Response
We are grateful for the suggestion that makes this study more friendly to the future readers! We have revised the title to the following one according to the suggestion!
Transcriptional dysregulations of seven non-differentially expressed genes as biomarkers of metastatic colon cancer
Reviewer 1, Question 2
ABSTRACT
- The English must be improved.
- Concerning the abbreviation of metastatic colon cancer (MCC) it is known to be mCC.
Response
We appreciate the suggestions! We have sought the English language editing service of LetPub, and comprehensively edited the whole manuscript. The editing certificate is available at:
It is worth noting that the title has been changed according to the reviewing suggestion, and the certificate shows the original title.
We have also gone through the whole manuscript to change the abbreviation of metastatic colon cancer from “MCC” to “mCC”.
Reviewer 1, Question 3
Keywords
- Did the authors mean differential expression by undifferential expression if so it must be corrected if not it shouldn’t be in the keywords as it is not a common scientific term it can be changed to gene expression as well.
Response
Thanks for this important suggestion to make the keywords more explicitly meaningful! We have changed the last keyword to “gene expression”.
Reviewer 1, Question 4
INTRODUCTION
- In this sentence of the manuscript, “Gut microbiome and carcinoembryonic antigen (CAE) have gradually become targeted CC biomarkers of a sub-population of the CC patients” the authors did not present the results of the references they listed first microbiome profile is still not a biomarker for CC and not used in diagnostic at all, on the other hand, CAE is general serum marker for most of the solid tumors in CC case clinicians use CEA with CA19 levels for CC diagnostic, therapy response and follow up.
Response
Thanks for this insightful comment! We have rephrased the sentence in the Introduction to the following one:
Carcinoembryonic antigen (CEA) serves as a general serum biomarker for multiple solid tumors, and its combination with the levels of carbohydrate antigen 19-9 (CA19-9) has been used for the clinical diagnosis, therapy response and prognosis follow-up decision of CC patients [9,10].
The supporting references have also been updated as:
- Vukobrat-Bijedic Z, Husic-Selimovic A, Sofic A, et al. Cancer Antigens (CEA and CA 19-9) as Markers of Advanced Stage of Colorectal Carcinoma. Med Arch, 2013, 67(6): 397-401, doi:10.5455/medarh.2013.67.397-401.
- Sreedhar, R., Jajoo, S., Yeola, M., Lamture, Y., & Tote, D. (2020). Role of Tumour Markers CEA and CA19-9 in Colorectal Cancer. J Evolution Med Dent Sci, 9, 3483-3488, doi: 10.14260/jemds/2020/762.
Reviewer 1, Question 5
INTRODUCTION
- Authors must define dark biomarkers in the introduction part and from where the nomenclature was given.
Response
We are sorry that we did not explicitly define the term “dark biomarker”. We have added the explicit definition of the term “dark biomarker” in the last paragraph of the Introduction section. In order to make the future readers easy to know the mearning of this term, we also added this definition in the Abstract.
Reviewer 1, Question 6
INTRODUCTION
- English must be improved.
Response
We appreciate this suggestion! We have sought the English language editing service of LetPub, and comprehensively edited the whole manuscript. The editing certificate is available at:
Reviewer 1, Question 7
MATERIALS AND METHODS
- The last date of any database and webpage access must be added with the link.
Response
We appreciate this suggestion! We have added the suggested information for each database and webpage in the manuscript, including GEO, AnimalTFDB, STRING, GeneCards, PubMed, UniProtKB, and LncBook.
Reviewer 1, Question 8
MATERIALS AND METHODS
- A detailed description of the statistical model used to analyze the data must be added also the software used.
Response
Thanks for the suggestion! We have provided the detailed descriptions of the statistical models and softwares used to analyze the data of Figures 2-4, and the “Design of experiment” section.
Reviewer 1, Question 9
RESULTS & DISCUSSION
- Figures quality and colors can be improved.
Response
We have redrawn all the Figures with larger font sizes and better color settings.
Reviewer 1, Question 10
RESULTS & DISCUSSION
- Most of the paragraph “Detection of MCC dark biomarkers” presents methods and the study design which must be switched to the methods section.
Response
Thanks for the suggestion! We have relocated this sub-section to the methods section.
Reviewer 1, Question 11
RESULTS & DISCUSSION
- There is no clear scientific message between involved genes in mCC and the transcription factors the pathway presented in the results hasn’t been discussed on the prognostic value of mCC and compared to other similar findings.
Response
We appreciate this insightful comment! We have added the detailed discussions about the suggested issues in the manuscript. In summary, the methodology limitation makes it difficult to directly evaluate the regulatory contributions of the individual TFs and to interpret the scientific messages and insights based on the models. We plan to explore the explainable artificial intelligence-based regression algorithms and ablation experiment-based evaluations to interpret the scientific messages and insights conveyed from the TFs to their regulatory gene in the future work.
We have also added the discussions of the pathways for their associations with mCC supported by the literature in the “mCC association with dark biomarkers based on the protein–protein interaction network” section. In summary, some of the identified dark biomarkers may exert their functions in mCC via their encoded or interacting proteins, although all these dark biomarkers do not show expression associations with mCC.
Reviewer 1, Question 12
RESULTS & DISCUSSION
- Results and discussion must be reorganized depending on the purpose of the study.
Response
We appreciate the insightful suggestions of this reviewer! According to the suggestions, we have relocated the “Detection of mCC dark biomarkers” sub-section to the methods section, and switched the orders of the two sub-sections “Many lncRNAs overlap with dark biomarkers” and “The protein level of YTHDC2 is associated with mCC”.
In summary, the first two sub-sections of the “Results and discussion” section described the detection procedure of the dark biomarkers. The third sub-section “Many lncRNAs overlap with dark biomarkers” discussed the potential reason for the non-differential expressions of these dark biomarkers. The last two sub-sections discussed that these dark biomarker genes might have got invovled in mCC via their encoded and interacting proteins.
Reviewer 1, Question 13
CONCLUSION
It must be improved we usually present in the conclusion section a short summary of the study funding in this manuscript it will be better if the authors can make their conclusion shorter by only referring to their important results.
Response
We appreciate this suggestion! We have removed the first paragraph on the background. The current “Conclusion” section consists of the first pagraph on the result summary of the study, the second paragraph on the possible experimental validation technologies, and the third paragraph on the methodology limitation to be resolved in the future work.
Reviewer 1, Question 14
Comments on the Quality of English Language
Authors must have a native English speaker better if not a scientist to review the manuscript or submit it for online English editing services to improve the quality of the manuscript and give better scientific soundness.
Response
We appreciate this suggestion! We have sought the English language editing service of LetPub, and comprehensively edited the whole manuscript. The editing certificate is available at:
Reviewer 2 Report
The study authors explicitly state the aim and objectives of the research.
There are a types of biomarkers that can determine the behavior of cancer cells. The topic is of broad interest without question to a wide audience.
Author Response
May 18, 2023
Dear Editor of Genes,
Please find the revised manuscript “Transcriptional dysregulations of seven non-differentially ex-pressed genes as biomarkers of metastatic colon cancer” submitted for the journal Genes. The original submission ID is genes-2363380, and the title has been changed according to the reviewing suggestion. The manuscript has been revised and the point-by-point responses are provided to all the reviewing comments from the reviewers. All the revisions are highlighted in red in the revised manuscript.
We really appreciate the insightful comments from the two anonymous reviewers that have help improve the manuscript a lot. We indicate our appreciation in the section Acknowledgements.
We hope that our revisions satisfied the requests of the editor and anonymous reviewers.
Thank you for giving us the revising opportunity to improve our manuscript!
Sincerely,
Fengfeng Zhou, Ph.D., Professor of Health Informatics
College of Computer Science and Technology, Jilin University.
Email: FengfengZhou@gmail.com or ffzhou@jlu.edu.cn
Reviewer 2, Question 1
Comments and Suggestions for Authors
The study authors explicitly state the aim and objectives of the research.
There are a types of biomarkers that can determine the behavior of cancer cells. The topic is of broad interest without question to a wide audience.
Response
We appreciate the positive comments from this reviewer! We have extensively revised and edited the manuscript according to the reviewing comments.
Round 2
Reviewer 1 Report
All of my comments were taken in confederation and the manuscript has been improved.